# Contrasting distinct structured views to learn sentence embeddings

## Abstract

We propose a self-supervised method that builds sentence embeddings from the combination of diverse explicit syntactic structures of a sentence. We assume structure is crucial to build consistent representations as we expect sentence meaning to be a function from both syntax and semantic aspects. In this perspective, we hypothesize that some linguistic representations might be better adapted given the considered task or sentence. We, therefore, propose to jointly learn individual representation functions for different syntactic frameworks. Again, by hypothesis, all such functions should encode similar semantic information differently and consequently, be complementary for building better sentential semantic embeddings. To assess such hypothesis, we propose an original contrastive multi-view framework that induces an explicit interaction between models during the training phase. We make experiments combining various structures such as dependency, constituency, or sequential schemes. We evaluate our method on standard sentence embedding benchmarks. Our results outperform comparable methods on several tasks.

## 1 Introduction

We propose a self-supervised method that builds sentence embeddings from the combination of diverse explicit syntactic structures. Such a method aims at improving the ability of models to perform compositional knowledge. In particular, we evaluate the embedding potential to solve downstream tasks.

Building generic sentence embeddings remains an open question. Many training methods have been explored: generating past and previous sentences (Kiros et al., 2015; Hill et al., 2016), discriminating context sentences (Logeswaran & Lee, 2018), predicting specific relations between pairs of sentences (Conneau et al., 2017; Nie et al., 2019). While all these methods propose efficient training objectives, they all rely on a similar RNN as encoder architecture. Nonetheless, model architectures have been subject to extensive work as well (Tai et al., 2015; Zhao et al., 2015; Arora et al., 2017; Lin et al., 2017), and in supervised frameworks, many encoder structures outperform standard RNN networks.

We hypothesize structure is a crucial element to perform compositional knowledge. In particular, the heterogeneity of performances given models and tasks makes us assume that some structures may be better adapted for a given example or task. Therefore, combining diverse structures should be more robust for tasks requiring complex word composition to derive their meaning. Hence, we aim here to evaluate the potential benefit from interactions between pairs of encoders. In particular, we propose a training method for which distinct encoders are learned jointly. We conjecture this association might improve our embeddings' power of generalization and propose an experimental setup to corroborate our hypothesis.

We take inspiration from multi-view learning, which is successfully applied in a variety of domains. In such a framework, the model learns representations by aligning separate observations of the same object. Traditionally, views are issued from a complementary natural perception of the data. For example, a picture and a sound recording of a dog. However, it can be extended to any pair of samples that share similar semantic content, such as the translation of the same sentence in two different languages. The definition can be extended to *synthetic* views, which are derived from the same unimodal data. In our case, we derived multiple *views* from a single sentence by pairing it with

a distinct syntactic framework. We illustrated in Figure 2, two *views* derived from the same input sentence by applying respectively a constituent or dependency parser.

As proposed in image processing (Tian et al., 2019; Bachman et al., 2019), we propose to align the different views using a contrastive learning framework. Indeed, contrastive learning is broadly used in NLP Mikolov et al. (2013b;a); Logeswaran & Lee (2018). We proposed to enhance the sentence embedding framework proposed in Logeswaran & Lee (2018) with a multi-view paradigm. As detailed in Section 2, composing multiple views has demonstrated its effectiveness in many NLP applications. However, as far as we are aware, combining distinct structured models to build standalone embeddings has not yet be explored. Nevertheless, this paradigm benefits from several structural advantages: as already mentioned, it pairs nicely with contrastive learning. It might thus be trained in a self-supervised manner that does not require data annotation. Moreover, contrary to models presented in Section 2, our method is not specific to a certain kind of encoder architecture. It does not require, for example, the use of attention layers or tree-structured models. More generally, it could be extended to any notion of view, even in other domains than language processing. Our setup could therefore be extended with any encoding function. Finally, our training method induces an interaction between models during inference and, paramountly, during the training phase.

Our paper is organized as follows: we detail our contrastive multi-view framework in Section 3. In Section 4, we propose an evaluation of our framework on standard evaluation benchmarks and propose qualitative analysis from our embeddings.

## 2    RELATED WORK

Multi-view is effectively used in a broad variety of domains. For image processing, some methods aim to learn representations by filling the missing part of an image or solving jigsaw puzzles. For video, Tian et al. (2019) propose to build image tuples using video frames and flow. For audio, van den Oord et al. (2018) maximize the mutual information between the embedding of the signal at different time steps.

Regarding NLP, combining different structural views has already been proven to be successful. Kong & Zhou (2011) provide a heuristic to combine dependency and constituency analysis for coreference resolution. Zhou et al. (2016); Ahmed et al. (2019) combine Tree LSTM and standard sequential LSTM with a cross-attention method and observe improvement on a semantic textual similarity task. Chen et al. (2017a) combine CNN and Tree LSTM using attention methods on a sentiment classification task, and CNN outperforms both Tree-LSTM and CNN separately. Finally, Chen et al. (2017b) combine sequential LSTM and Tree LSTM for natural language inference tasks. However, to our knowledge, combining distinct structured models in a contrastive learning setup was not attempted to build sentence embeddings.

## 3    METHOD

Given a sentence $s$, the model aims at discriminating the sentences $s^+$ in the neighborhood of $s$ from sentences $s^-$ outside of this neighborhood. This is contrastive learning (Section 3.1). The representation of each sentence is acquired by using multiple views (Section 3.2).

### 3.1    CONTRASTIVE LEARNING

Contrastive learning is successfully applied in a variety of domains including audio van den Oord et al. (2018), image (Wu et al., 2018; Tian et al., 2019), video or natural language processing for word embedding (Mikolov et al., 2013b) or sentence embedding (Logeswaran & Lee, 2018). Some mathematical foundations are detailed in Saunshi et al. (2019). The idea is to build a dataset such that each sample $x$ is combined with another sample $x^+$, which is somehow *close*. For word or sentence embeddings, the close samples are the words or the sentences appearing in the given textual context. For image processing, close samples might be two different parts of the same image. Systems are trained to bring close sentences together while dispersing negative examples.

In particular, a sentence embedding framework is proposed by Logeswaran & Lee (2018). The method takes inspiration from the distributional hypothesis successfully applied for word, but this

time, to identify context sentences. The network is trained using a contrastive method. Given a sentence $s$, a corresponding context sentence $s^+$ and a set of $K$ negative samples $s_1^- \cdots s_K^-$, the training objective is to maximize the probability of discriminate the correct sentence among negative samples: $p(s^+|s, s_1^- \cdots s_K^-)$.

The algorithm architecture used to estimate $p$ is close to *word2vec* (Mikolov et al., 2013b;a). Two sentences encoders $f$ and $g$ are defined and the conditional probability is estimated as follow:

$$p(s^+|s, s_1^- \cdots s_K^-) = \frac{e^{f(s)^T g(s^+)}}{e^{f(s)^T g(s^+)} + \sum_{i=1}^N e^{f(s)^T g(s_i^-)}}$$

At inference time, the sentence representation is obtained as the concatenation of the two encoders $f$ and $g$ such as $s \rightarrow [f(s); g(s)]$. In Logeswaran & Lee (2018), $f$ and $g$ are chosen identical and consist in two RNN. However, the authors observe that the encoders might learn redundant features. To limit this effect, they propose to use a distinct set of embeddings for each encoder.

We propose addressing this aspect by enhancing the method with a multi-view framework and using a distinct structured model for the encodes $f$ and $g$. We hypothesize that some structures may be better adapted for a given example or task. For example, Figure 2 illustrates that dependency parsing uses the verb "filled" as the root node. Whereas in constituency parsing, subject and verb are respectively, the right and left child from the root node. Therefore, the combination of different structures should be more robust for tasks requiring complex word composition and be less sensitive to lexical variations. Consequently, we propose a training procedure that allows the model to benefit from the interaction of various syntactic structures. The choice for the encoder architecture is detailed in the following section.

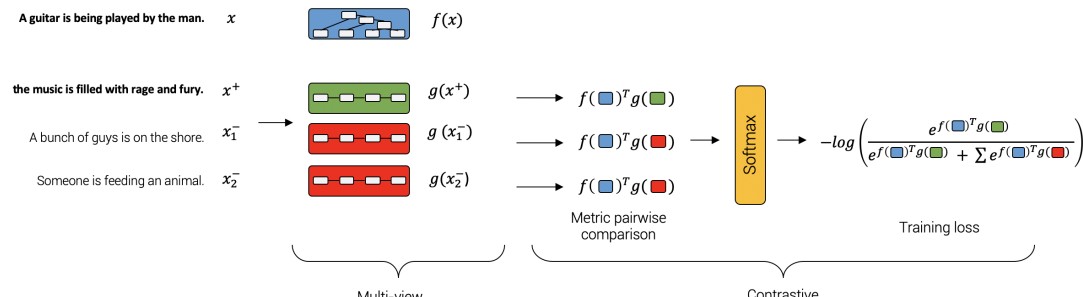

Figure 1: **Contrastive training method**. The objective is to reconstruct the storyline. Sentences are presented in their original order. Given an anchor sentence $x$, the algorithm should identify the context sentence $x^+$ out of negative samples $x_1^-, x_2^-$. Sentences are encoded using separate views, which are composed within a pairwise distance matrix. A softmax classifier outputs the probability of each sentence pair to share the same context. In practice, we consider two distance matrices given the view $f$ or $g$ used to encode the anchor sentence and the target samples. We estimate the final probability as the average of the two resulting probabilities.

## 3.2 LANGUAGE VIEWS

Multi-view aims as learning representations from data represented by multiple independent sets of features. The method is not specific for any particular nature of data and can be applied to a broad scale of domains, making it an efficient framework in self-supervised representation learning. As depicted in Section 1, we generalize the notion of view for a sentence as the application of a specific syntactic framework. For each view, we use an ad-hoc algorithm that maps the structured representation of the sentence into an embedding space.

For tree-based views, we consider both phrase structure trees and dependency trees. The phrase structure of a sentence is represented as nested multi-word constituents. The dependency tree represents the relationship between individual words. Although equivalences might be derived between

the two representations schemes, we hypothesize that, in our context, the corresponding sequence of operations might allow capturing rather distinct linguistic properties. The various models may, therefore, be complementary and their combination allows for more fine-grained analysis. Together with the trees, we consider the following views of a sentence.

**Bag Of Word (BOW)**  This setup does not assume any underlying structure. The sentence is modeled as an unordered set of words. The associated encoding method is a simple commutative sum operation of the word embeddings. In our case, vectors are initialized using GloVe vectors (Pennington et al., 2014) publicly available[1]. We used 300-dimensional word vectors trained on the common crawl dataset (840B tokens) with a vocabulary of 2.2M case sensitive words.

**Vanilla LSTM (SEQ)**  assumes a sequential structure where each word depends on the previous words in the sentence. The framework is a bidirectional sequential LSTM (Hochreiter & Schmidhuber, 1997). The concatenation of the forward and backward last hidden state of the model is used as sequence embedding.

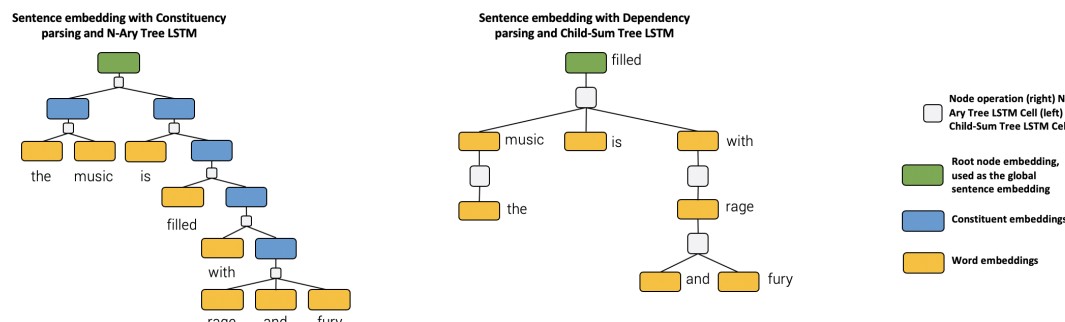

Figure 2: *(left)* The sentence is parsed in constituency and the tree is binarized. The application of the N-Ary Tree LSTM on the obtained structure is represented. *(right)* The sentence is parsed in dependency and a Child-Sum Tree LSTM model is recursively applied. This example illustrates the structural difference between these two views. Dependency parsing is articulated around the verb "filled", which is the root node. In constituency, subject and verb are connected through the root node. The two architectures differ as the N-Ary Tree LSTM is structured as a binary tree and differentiate the left and right children while the Child-Sum Tree LSTM might have an arbitrary number of unordered nodes.

**Dependency tree (DEP)**  In the dependency tree model, words are connected through dependency edges. A word might have an arbitrary number of dependants. As illustrated in Figure 2, the sentence can be represented as a tree where nodes correspond to words and edges indicate whether or not the words are connected in the dependency tree. In our case, the dependency tree is obtained using the deep biaffine parser from Dozat & Manning (2017)[2] For this view, we compute sentence embeddings with the Child-Sum Tree LSTM model described in Tai et al. (2015): Each node is assigned an embedding given its dependent with a recursive function. The recursive node function is derived from standard LSTM formulations but adapted for tree inputs. In particular, the hidden state is computed as the sum of all children hidden states:

$$\tilde{h}_j = \sum_{k \in C(j)} h_k \tag{1}$$

with $C(j)$, the set of children of node $j$. All equations are detailed in Tai et al. (2015). However, we slightly modify the computation of $\tilde{h}_j$ using Equation 2. As in Zhou et al. (2016), we propose to

---

[1] https://nlp.stanford.edu/projects/glove/

[2] We use an open-source implementation of the parser and replace the pos-tags features with features obtained with BERT. Therefore we do not need pos-tags annotations to parse our corpus: https://github.com/yzhangcs/biaffine-parser

compute $\tilde{h}_j$ as the weighted sum of children vectors in order to allow the model to filter semantically less relevant children.

$$\tilde{h}_j = \sum_{k \in C(j)} \alpha_{kj} h_k \tag{2}$$

The parameters $\alpha_{kj}$ are attention weights computed using a *soft attention layer*. Given a node $j$, we consider $h_1, h_2, \ldots, h_n$ the corresponding children hidden states. the soft attention layer produces a weight $\alpha_k$ for each child's hidden state. We did not use any external query to compute the attention but instead use a projection from the current node embedding. The attention mechanism is detailed in equations below:

$$q_j = W^{(q)} x_j + b^{(q)} \tag{3}$$

$$p_k = W^{(p)} h_k + b^{(p)} \tag{4}$$

$$a_{kj} = \frac{q_j \cdot p_k^\mathsf{T}}{\|q_j\|_2 \cdot \|p_k\|_2} \tag{5}$$

$$\alpha_{kj} = \mathrm{softmax}_k(a_{1j} \cdots a_{nj}) \tag{6}$$

The embedding at the root of the tree is used as the sentence embedding as the Tree LSTM model computes representations bottom up.

**Constituency tree** (CONST)    Constituent analysis describes the sentence as a nested multi-word structure. In this framework, words are grouped recursively in constituents. In the resulting tree, only leaf nodes correspond to words, while internal nodes encode recursively word sequences. The structure is obtained using the constituency neural parser from Kitaev & Klein (2018). The framework is associated with the N-Ary Tree LSTM, which is defined in Tai et al. (2015). Similarly to the original article, we binarize the trees to ensure that every node has exactly two dependents. The binarization is performed using a left markovization and unary productions are collapsed in a single node. Again the representation is computed bottom-up and the embedding of the tree root node is used as sentence embedding. The equations detailed in Tai et al. (2015) make the distinction between right and left nodes. Therefore we do not propose to enhance the original architecture with a weighted sum as on the DEP view.

## 4    EXPERIMENTS

### 4.1    TRAINING CONFIGURATION

We train our models[3] on the UMBC dataset[4] Han et al. (2013). We filter 40M sentences from the tokenized corpus. We build batches from successive sentences. Given a sentence in a batch, other sentences not in the context are considered as negatives samples as presented in Section 3.1.

For the vocabulary, we follow the setup proposed in Logeswaran & Lee (2018) and we train two models in each configuration. One initialized with pre-trained embedding vectors. The vectors are not updated during training and the vocabulary includes the top 2M cased words from the 300-dimensional GloVe embeddings Pennington et al. (2014). The other is limited 50K words initialized with a Xavier distribution and updated during training. For inference, the vocabulary is expanded to 2M words using the linear projection proposed in Logeswaran & Lee (2018); Kiros et al. (2015).

All models are trained using a batch size of 400 and the Adam optimizer with a $5e^{-4}$ learning rate. Regarding the infrastructure, we use a Nvidia GTX 1080 Ti GPU. All model weights are initialized with a Xavier distribution and biases set to 0. We do not apply any dropout. For the SEQ view, we use GRU cells instead of LSTM to match the implementation proposed in Logeswaran & Lee (2018) setup.

---

[3]Hyperparameters of the models such as the hidden size and the optimization procedure such as learning rate are fixed given literature on comparable work. In particular Tai et al. (2015); Logeswaran & Lee (2018) and are detailed in Appendix.

[4]The bookcorpus introduced in (Zhu et al., 2015) and traditionally used sentence embedding is no longer distributed. Therefore, we prefer a corpus freely available. The UMBC dataset is available at: `https://ebiquity.umbc.edu/blogger/2013/05/01/umbc-webbase-corpus-of-3b-english-words/`

## 4.2 Evaluation on downstream tasks

| Model | Dim | Hrs | MR | CR | SUBJ | MPQA | TREC | MRPC | | SICK-R | | MSE |
|---|---|---|---|---|---|---|---|---|---|---|---|---|
| | | | | | | | | Acc | F1 | $r$ | $\rho$ | |
| *Context sentences prediction* | | | | | | | | | | | | |
| FastSent | ≤ 500 | 2 | 70.8 | 78.4 | 88.7 | 80.6 | 76.8 | 72.2 | 80.3 | — | — | — |
| FastSent + AE | ≤ 500 | 2 | 71.8 | 76.7 | 88.8 | 81.5 | 80.4 | 71.2 | 79.1 | — | — | — |
| Skipthought | 4800 | 336 | 76.5 | 80.1 | 93.6 | 87.1 | 92.2 | 73.0 | 82.0 | 85.8 | 79.2 | 26.9 |
| Skipthought + LN | 4800 | 672 | 79.4 | 83.1 | 93.7 | 89.3 | — | — | — | 85.8 | 78.8 | 27.0 |
| Quickthoughts | 4800 | 11 | **80.4** | **85.2** | **93.9** | **89.4** | **92.8** | 76.9 | 84.0 | **86.8** | **80.1** | 25.6 |
| *Sentence relations prediction* | | | | | | | | | | | | |
| InferSent | 4096 | — | **81.1** | **86.3** | 92.4 | 90.2 | 88.2 | **76.2** | 83.1 | **88.4** | — | — |
| DisSent Books 5 | 4096 | — | 80.2 | 85.4 | 93.2 | 90.2 | 91.2 | 76.1 | — | 84.5 | — | — |
| DisSent Books 8 | 4096 | — | 79.8 | 85.0 | **93.4** | **90.5** | **93.0** | 76.1 | — | 85.4 | — | — |
| *Pre-trained transformers* | | | | | | | | | | | | |
| BERT-base [CLS] | 768 | 96 | 78.7 | 84.9 | 94.2 | 88.2 | **91.4** | 71.1 | — | 75.7† | — | — |
| BERT-base [NLI] | 768 | 96 | **83.6** | **89.4** | 94.4 | 89.9 | 89.6 | **76.0** | — | 84.4† | — | — |
| ***Our models (GloVe & Pretrained Embeddings)*** | | | | | | | | | | | | |
| SEQ, CONST† | 4800 | 41 | 79.8 | 82.9 | 94.6 | 88.5 | 90.4 | 76.4 | 83.7 | 86.1 | 78.9 | 26.3 |
| DEP, SEQ† | 4800 | 27 | 79.7 | 82.2 | 94.4 | 88.6 | 91.0 | **77.9** | **84.4** | 86.6 | 79.8 | 25.5 |
| DEP, CONST† | 4800 | 39 | **80.7** | **83.6** | **94.9** | 89.2 | 92.6 | 76.8 | 83.6 | **87.0** | **80.3** | **24.8** |

Table 1: **SentEval Task Results Using Fixed Sentence Encoder.** We divided the table into sections. The first range of models is directly comparable to our model as the training objective is to identify context sentences. The second section objective is to identify the correct relationship between a pair of sentences. The third section reports pre-trained transformers based-models. The last section reports the results from our models. FastSent is reported from Hill et al. (2016). Skipthoughts results from Kiros et al. (2015) Skipthoughts + LN which includes layer normalization method from Ba et al. (2016). We considered the Quickthoughts results Logeswaran & Lee (2018) with a pre-training on the bookcorpus dataset. DisSent and Infersent are reported from Nie et al. (2019) and Conneau et al. (2017) respectively. Pre-trained transformers results are reported from Reimers & Gurevych (2019). The **Hrs** column indicates indicative training time, the **Dim** column corresponds to the sentence embedding dimension. † indicates models that we had to re-train. Best results in each section are shown in **bold**, best results overall are underlined. Performance for **SICK-R** results are reported by convention as $\rho$ and $r \times 100$.

As usual for models aiming to build generic sentence embeddings (Kiros et al., 2015; Hill et al., 2016; Arora et al., 2017; Conneau et al., 2017; Logeswaran & Lee, 2018; Nie et al., 2019), we use the SentEval benchmark. SentEval is specifically designed to assess the quality of the embeddings themselves rather than the quality of a model specifically targeting a downstream task, as is the case for the GLUE and SuperGLue benchmarks (Wang et al., 2019b;a). Indeed, the evaluation protocol prevents for fine-tuning the model during inference and the architecture to tackle the downstream tasks is kept minimal. Moreover, the embedding as kept identical for all tasks, thus assessing their properties of generalization.

Therefore, classification tasks from the SentEval benchmark are usually used for evaluation of sentence representations[5] (Conneau & Kiela, 2018): the tasks include sentiment and subjectivity analysis (**MR, CR, SUBJ, MPQA**), question type classification (**TREC**), paraphrase identification (**MRPC**) and semantic relatedness (**SICK-R**). Contrasting the results of our models on this set of tasks will help to better understand its properties.

We use either the pre-defined train/dev/test splits or perform a 10-fold cross-validation. We follow the linear evaluation protocol of Kiros et al. (2015), where a logistic regression or softmax classifier

---

[5]SentEval tool and data are freely available: `https://github.com/facebookresearch/SentEval`

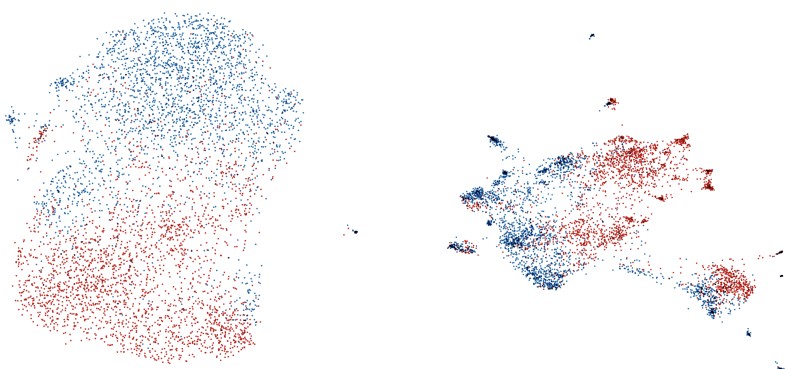

Figure 3: Projection of the embeddings from the **SUBJ** task. *(left)* The DEP, CONST model is used *(right)* We train a Quick-thought model using scripts from Logeswaran & Lee (2018)[7] on the UMBC dataset. Both dimension reductions are performed using the UMAP algorithm. In both cases, samples appear well separated given their labels.

is trained on top of sentence representations. The dev set is used for choosing the regularization parameter and results are reported on the test set.

### 4.3 EVALUATION ON DOWNSTREAM TASKS

We compare the properties of distinct views combination on downstream tasks and report the results with respect to comparable state of the art methods in Table 1. The first set of methods (*Context sentences prediction*) relies on a distributional hypothesis: models are trained to reconstruct books storyline.

The second set of models (*Sentence relations prediction*) is pre-trained on a supervised task. Infersent Conneau et al. (2017) is trained on the SNLI dataset, which proposes to predict the entailment relation between two sentences. DisSent Nie et al. (2019) proposes a generalization of the method and builds a corpus of sentence pairs with more possible relations between them. Finally, we include models relying on transformer architectures (Pre-trained transformers) for comparison. In particular, BERT-base model and a BERT-model fine-tuned on the SNLI dataset Reimers & Gurevych (2019). In Table 1, we observe that our models expressing a combination of views such as (DEP, SEQ) or (DEP, CONST) give better results than the use of the same view (SEQ, SEQ) used in Quick-Thought model. It seems that the entanglement of views benefits the sentence embedding properties. In particular, we obtain state-of-the-art results for almost every metric from **MRPC** and **SICK-R** tasks, which focus on paraphrase identification. For the **MRPC** task, we gain a full point in accuracy and outperform BERT models. We hypothesize structure is important for achieving this task, especially as the dataset is composed of rather long sentences. The **SICK-R** dataset is structurally designed to discriminate models that rely on compositional operations.

This also explains the score improvement on this task. Tasks such as **MR**, **CR** or **MPQA** consist in sentiment or subjectivity analysis. We hypothesize that our models are less relevant in this case: such tasks are less sensitive to structure and depend more on individual word or lexical variation.

We finally observe our setup is competitive with models trained on broader datasets. Indeed, we use the publicly available UMBC corpus and limited the size to 40M sentences. In comparison, the BookCorpus used in Kiros et al. (2015); Logeswaran & Lee (2018) consists in 74M sentences.

### 4.4 QUALTITATIVE RESULTS

We analyze the embeddings from a qualitative perspective and explore the sentences from the **SICK-R** test set. We retrieved the closest neighbors using cosine distance. We compare the results with the Quick-thought model. We illustrated in Table 2 a panel of examples presenting interesting linguistic properties. Models seem somehow robust to adjective expansions illustrated in the first examples. Indeed, the closest expression from *"A black bird "* is *"A bird , which is black"*. However the second

| Encoder | Query and two closest sentences | Cosine distance |
|---|---|---|
| | *A black bird is sitting on a dead tree* | |
| DEP, CONST | A bird , which is black , is sitting on a dead tree | 0.118 |
| | A dead bird is near a black man sitting on a tree | 0.139 |
| CONST, SEQ | A bird , which is black , is sitting on a dead tree | 0.118 |
| | The black bird is sitting in a leafless tree | 0.143 |
| Quickthoughts | A bird , which is black , is sitting on a dead tree | 0.172 |
| | A dead bird is near a black man sitting on a tree | 0.172 |
| | *Rugby is being played by some men* | |
| DEP, CONST | Rugby players are tackling each other | 0.381 |
| | Some men are playing rugby | 0.392 |
| CONST, SEQ | Guitar is being played by two men | 0.401 |
| | Rugby players are tackling each other | 0.403 |
| Quickthoughts | Guitar is being played by two men | 0.455 |
| | Rugby players are tackling each other | 0.462 |
| | *A crowd of people is near the water* | |
| DEP, CONST | A crowd of people is far from the water | 0.079 |
| | A group of people is near the ocean | 0.356 |
| CONST, SEQ | A crowd of people is far from the water | 0.063 |
| | A man is coming out of the water | 0.313 |
| Quickthoughts | A crowd of people is far from the water | 0.067 |
| | Two people are wading through the water | 0.388 |

Table 2: **A qualitative exploration of the sentence embedding space** We embed the sentences from the **SICK-R** test set. Given a query sentence, we retrieve the closest two sentences from the dataset using cosine distance. We compare the results of the semantic search using distinct views or single views combinations.

retrieved sentence is semantically correct for the CONST, SEQ association only. Quick-thought and DEP, CONST present a weakness toward word scrambling for this specific example. We investigate passive forms in the second example. The CONST, SEQ and Quickthough models seem to attach to much weight to the sentence syntax rather than the semantic. This time the association of DEP and CONST views retrieve to corresponding active sentences. Finally, we observe how models behave when facing numeric information. Interestingly Quickthoughts and DEP, CONST are able to bring together "*crowd*" and "*group*" notions.

From a graphic perspective, we projected in two dimensions the sentences from the **SUBJ** task, for which we obtained state-of-the-art results. We use the UMAP algorithm for dimensionality reduction and compare our multi-view setup with the Quick-thought model. The projection is illustrated in Figure 3. While the Figure does not reveal any critical distinction between models, samples appear well separated in both cases.

## 5 CONCLUSION AND FUTURE WORK

Inspired from linguistic insights and work on supervised learning, we hypothesize that structure is a central element to build sentence embedding. We propose in Section 3 an original contrastive multi-view framework that aims to build embeddings from the interaction of various structured models.

In Section 4, we proposed to assess the quality of our embeddings and use them as a feature for the dedicated SentEval benchmark. We obtain state-of-the-art results on tasks which are expected, by hypothesis, to be more sensitive to sentence structure. Exploring our representation space from a qualitative perspective, we observe the chosen pair of views might affect the reaction to linguistic perturbations such as passive forms or adjective expansions. Some view pairs appear indeed more robust for some specific examples.

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
