# OpenReview forum: "Contrasting distinct structured views to learn sentence embeddings"
_ICLR.cc/2021/Conference — Reject_

### Official Review · AnonReviewer1 · 2020-10-26

**Rating:** 5
**Confidence:** 4

**Review:**


Thank the authors for showing their effort in revising the paper.
Some of my concerns have been solved thanks to the rebuttal.

- - - - - - - - - - - - - - - - - - - - - - - - - - - - - - - - - - - - - - - - - - - - - - - - - - - - - - - - - - - - - - - - - - - - - - - - - - - - - - - - - - - - - - - - - - - - - - - - - - - - - - - -

#### Summary

- This paper proposes a self-supervised (multi-view learning) method that constructs sentence embeddings by exploiting different types of sentence encoders, including tree encoders based on constituency and dependency trees. The intuition is that the integration of respective representations from disparate (linguistically-informed) encoders is effective since each linguistic formalism can provide different linguistic views. In experiments, the proposed method was tested on the SentEval benchmark and demonstrated its effectiveness compared to other baselines. The authors also conducted some qualitative analyses to reveal the inner working of their method.

#### Pros (Reasons to Accept)

- The introduction of (constituency and dependency) tree encoders to sentence embedding construction from the perspective of multi-view learning.

#### Cons (Reasons to Reject)

- Weak novelty: This work attempts to improve upon Logeswaran & Lee (2018) (so-called 'Quickthoughts') by applying tree (and some other) encoders in addition to sequential RNN encoders. However, it is hard to find a novel technical contribution that only appears in this paper, except that the authors demonstrated that the simple combination of two well-known concepts (i.e., Quickthoughts and tree encoders) is empirically effective for building sentence embedding. Even though the results presented in the paper is interesting, in my opinion, there should be a stronger contribution that leads to this paper to be unique and informative.
- Doubts on the practical usefulness of the proposed sentence embeddings: Compared against typical frameworks that accept just plain sentences as input, the proposed framework requires input sentences to be parsed by external parsers, which might be problematic: Specifically, (1) the reliance on external parsers may cause unexpected error propagations from the parsers, considering that existing parsers are not perfect (in spite of recent improvements in developing good parsers). Moreover, reliable parsers are only available for a few resource-rich languages such as English. Therefore, some concerns about how to parse input sentences appropriately in different (or harsh) conditions should be (at least, briefly) covered in the paper. (2) As tree encoders are hard to be optimized with batch computations, there must be a decrease in inference speed if a model includes tree encoders. Considering this problem, the paper would be much persuasive when the inference speed of each compared model (in addition to its training time) is provided in experiments.

#### Comments

- I find myself being hard to believe that the comparison shown in Table 1 is fair enough, considering the following concerns. First, there are two factors that may impact a model's performance---(A) the architecture of the model and (B) the data used for training the model. Even though (A) is controlled in the presented experiments, (B) is not. In other words, readers cannot infer whether the performacne of a model is due to its architecture or its training data. I understand it's difficult and time-consuming to re-train all baselines. Nonetheless, the proposed model should be at least compared with the Quickthoughts model, which is the foundation of this paper, in the same condition for a fair comparison.
- To make the paper more convincing, I recommend that the authors test the BERT model in the condition where the dimensionality of the BERT embedding is expanded to larger numbers, making it comparable to those of other models.
One possible solution is to follow the method proposed in John Wieting & Douwe Kiela (2019).
For more details, please refer to John Wieting & Douwe Kiela, No Training Required: Exploring Random Encoders for Sentence Classification (ICLR 2019).

#### Misc.

- (Typo) Section 3.1. encodes -> encoders

---

### Official Review · AnonReviewer6 · 2020-11-04
**Is explicit syntax really necessary?**

**Rating:** 3
**Confidence:** 4

**Review:**

The authors introduce a pretrianing paradigm based on contrastive learning between multiple syntactic views of the same sentence. The method maximizes representations between different setence encoders when given the same sentence, and minimize the similarity to all other sentence repre sentations. The results on the infersent benchmark show competitive performance of the approach when compared to non-syntactic pretraining methods.

There are a couple of concerns I have with the paper and some questions I will elaborate on in detail. In particular, this paper doesn't convince me that incorporating explicit structure is i) desirable and ii) necessary to achieve good results on the chosen benchmarks. I also have doubts about the comparison and the bechmark in general, especially given the fact that random encoders (wo/ explicit syntax) on pretrained word embeddings can do a pretty good job on those [2].

Therefore, at the current state of the paper I cannot recommend acceptance.

Detailed comments:

1) Are explicit, human designed syntax frameworks really needed? The authors argue that explicit syntax can help generalization, which I could maybe agree with, if trained parsers were perfect. Since they themselves are far from that and regularly fail on complicated sentence structures, I don't believe that this will assist in generalization. Rather the errors in parsing can lead to more systematic failures that the models might have a harder time to correct downstream. Prior work shows that Transformers trained on a (masked) LM objective learn about sentence structure implicitly [1]. In fact, although the authors "hypothesize" that explicit structure can help -- which they do not properly ablate in my opinion (see 2-3) --, the results in the paper show no clear advantages on their benchmarks when comparing to BERT. The increased complexity of the method (ie., relying on other trained models) should be warrented by showing a clear advantage in doing so.

2) Fair comparisons: The models in the paper are trained on another pretraining corpus. Other baselines should have been trained on this corpus to have an apples to apples comparison. Furthermore, BERT models use much lower dimensional sentence embeddings which typically hurts performance. A way to make comparison fairer would be to apply a random projection of the 768 features to 4800 features. Note that this can have a dramatic impact even for BoW models on these benchmarks [2]. I also wonder how well a randomly initialized model would perform, that is, how much does the pretraining actually help? [2] shows that it might not be required at all. I think given the inductive bias coming from the parser and the use of TreeLSTMs, this effect might even be greater, because much of the information about the structure sentence is given apriori, which might even help random models to achieve better performance.

3) What can we expect the pretraining to learn? For instance, wouldn't a satisfactory solution to the pretraining task for the 2 encoders to simply learn a bag-of-words representation? I believe that the large majority of the setences could easily be distinguished by that. Just memorize the exact words that occured in the sentence. I am not convinced from the paper that the model learns anything more semantic than that. More rigorous ablations are required to show the benefit of the pretraining method.


My personal opinion on introducing explicit syntax into neural nets (note that this will not be part of my decision):

Given all the evidence we have so far on pretraining LMs using generic model such as transformers, I just don't see any reason why we should still try to explicitly fit our potentially faulty and biased syntactic frameworks into neural networks when more generic models (such as transformers) can learn structure directly from the data. Our linguistic frameworks can still be super useful for understanding and systematically testing our models, but I think we should refrain from  introducing our potentially limited understanding of language into those models. A random thought: How would pretrained parsers help a model on twitter like text or child speech?


[1] Jawahar et al., What Does BERT Learn about the Structure of Language? ACL, 2019.

[2] Wieting et al., NO TRAINING REQUIRED: EXPLORING RANDOM ENCODERS FOR SENTENCE CLASSIFICATION. ICLR 2019.

---

### Official Review · AnonReviewer5 · 2020-11-05
**lack of justification and not convincing results**

**Rating:** 4
**Confidence:** 4

**Review:**

The paper proposed an ensemble of sentence encoders (RNN, BoW, tree-based) that is trained with contrastive loss. The proposed approach is a simple extension of the work of Logeswaran & Lee (2018). Overall, I think the paper has several major weaknesses.

First, the results on SentEval are not convincing in comparison to other general purpose methods (i.e., BERT or QuickThought) which do not require a parsed tree during training and inference. The results also do not justify the use of additional resources (dependency and constituency parsers) as it complicated the models and is not applicable for many other languages where parsers are not available or their performances are still far from usable. Note that, there are several pre-trained models after BERT (e.g., XLNet, XLM-R, ALBERT,...) which can be used to obtain sentence embeddings. I think the authors should also compare the results with those models.

Second, SentEval framework suffers from some drawbacks [1]. Using it alone is insufficient to draw any meaningful conclusion about sentence embeddings. I think that the authors could have used probing tasks [2] to evaluate their sentence embeddings with respect to linguistic properties.

Third, while the paper claims posed a hypothesis that structure is crucial to build consistent representations, this is not shown in the paper. I think that the paper lacks some analysis/examples  to show in which cases modelling structure of the sentence explicitly is necessary. Other models that use Transformers or RNN exploit structure in language implicitly in one way or another. For example, Hewitt and Manning shows that syntax is embedded in deep models (i.e., BERT). Thus It’s not clear to me what is the advantage of the proposed approach in comparison to other models.

Comments for the authors:  I think that the comparison with previous work in table 1 is not meaningful since the authors used another dataset for training.



[1] [Pitfalls in the Evaluation of Sentence Embeddings](https://www.aclweb.org/anthology/W19-4308/). Steffen Eger, Andreas Rücklé, Iryna Gurevych. RepL4NLP-2019

[2] [What you can cram into a single $&!#* vector: Probing sentence embeddings for linguistic properties](https://www.aclweb.org/anthology/P18-1198/). Alexis Conneau, German Kruszewski, Guillaume Lample, Loïc Barrault, Marco Baroni. ACL 2018

[3] [A Structural Probe for Finding Syntax in Word Representations](https://www.aclweb.org/anthology/N19-1419/). John Hewitt Christopher D. Manning. NAACL 2019

---

### Author Response · Authors · 2020-11-19
**Response to reviews**

Dear reviewers, thanks for your detailed reviews and comments. We appreciate the time you spent challenging our work. We notice some specific points are shared across your reviews and we focused on these specific aspects.

**Regarding the comparison with previous work in Table 1.** We have indeed chosen to make use of a distinct corpus as the BookCorpus dataset is no longer distributed for copyright reasons. We have run QuickThought scripts using our dataset based on the UMBC corpus and compared both setups below:

| &nbsp;&nbsp;task&nbsp;&nbsp;                                            | &nbsp;&nbsp;Dim&nbsp;&nbsp;  | &nbsp;&nbsp;MR&nbsp;&nbsp;   | &nbsp;&nbsp;CR&nbsp;&nbsp;   | &nbsp;&nbsp;SUBJ&nbsp;&nbsp; | &nbsp;&nbsp;MPQA&nbsp;&nbsp; | &nbsp;&nbsp;TREC&nbsp;&nbsp; | &nbsp;&nbsp;MRPC&nbsp;&nbsp; | &nbsp;&nbsp;SICK-R&nbsp;&nbsp; |
|:--------------------------------------:|:--------|:------:|:-----:|:---------:|:------:|:-------:|:--------:|:---------:|
| Quickthoughts&nbsp;&nbsp;                         | 4800 | 80.4 | 85.2 | 93.9    | 89.4 | 92.8  | 76.9    | 0.87     |
| Quickthoughts (UMCB 45M)&nbsp;&nbsp; | 4800 | 80.9 | 84.4 | 95.1    | 88.9 | 92.2   | 75.8    | 0.86      |

From our point of view, the results are relatively close. Indeed, for the majority of tasks, the use of our dataset penalizes the results. Our corpus is indeed restricted to 45M sentences, in comparison with 74M for the Bookcorpus. Given the dataset size and the SentEval results, we have considered the comparison holds.

Second, you expressed concerns about potential **pitfalls regarding the evaluation of sentence embeddings methods**. Given your references, sentence embedding evaluation methods may and suffer from several biases.

Regarding the dependency on the embedding size, we implemented your proposition to project Bert CLS token embedding using a random matrix initialized with a glorot distribution. This setup expands bert embedding into 4096 dimensions. However, we did not observe a significant gap between the results on SentEval:

| &nbsp;&nbsp;task&nbsp;&nbsp;                                            | &nbsp;&nbsp;Dim&nbsp;&nbsp;  | &nbsp;&nbsp;MR&nbsp;&nbsp;   | &nbsp;&nbsp;CR&nbsp;&nbsp;   | &nbsp;&nbsp;SUBJ&nbsp;&nbsp; | &nbsp;&nbsp;MPQA&nbsp;&nbsp; | &nbsp;&nbsp;TREC&nbsp;&nbsp; | &nbsp;&nbsp;MRPC&nbsp;&nbsp; | &nbsp;&nbsp;SICK-R&nbsp;&nbsp; |
|:--------------------------------:|:----:|:----:|:----:|:----:|:----:|:----:|:----:|:------:|
| BERT-\[CLS\]&nbsp;&nbsp;                     | 768  | 77.3 | 81.3 | 92.7 | 85.0 | 80.2 | 69.9 | 0.61   |
| BERT-\[CLS\] + random projection&nbsp;&nbsp; | 4096 | 77.1 | 82.6 | 93.1 | 85.9 | 80.8 | 71.3 | 0.71   |

Regarding the effect of randomly initialized encoders [2], we reported the results  on SentEval below. Although randomly structured encoders achieve surprisingly good results, they still below the one obtained with pre-training.

| &nbsp;&nbsp;task&nbsp;&nbsp;                                            | &nbsp;&nbsp;Dim&nbsp;&nbsp;  | &nbsp;&nbsp;MR&nbsp;&nbsp;   | &nbsp;&nbsp;CR&nbsp;&nbsp;   | &nbsp;&nbsp;SUBJ&nbsp;&nbsp; | &nbsp;&nbsp;MPQA&nbsp;&nbsp; | &nbsp;&nbsp;TREC&nbsp;&nbsp; | &nbsp;&nbsp;MRPC&nbsp;&nbsp; | &nbsp;&nbsp;SICK-R&nbsp;&nbsp; |
|:------------------------:|:----:|:----:|:----:|:----:|:----:|:----:|:----:|:------:|
| Rand LSTM&nbsp;&nbsp;                | 4800 | 77.2 | 78.7 | 91.9 | 87.9 | 86.5 | 74.1 | 0.86   |
| Quickthoughts&nbsp;&nbsp;             | 4800 | 80.4 | 85.2 | 93.9 | 89.4 | 92.8 | 76.9 | 0.87   |
| DEP / CONST&nbsp;&nbsp;               | 4800 | 80.7 | 83.6 | 94.9 | 89.2 | 92.6 | 76.8 | 0.87   |

Finally, we agree **our method requires additional data**, namely external parsers to provide the structure. From our point of view, relevant parsers are available for many languages. For instance, [3] and [4] gather efficient parsers in English, Chinese, Danish, French, Dutch, German and Greek. Obviously, not all languages are available. However, pre-trained models such as Bert also require a large amount of data to be trained, which may not be available in low resource languages.

Regarding the **inference speed**, The constituency parser is the bottleneck in this case and parse around 500 sentences/second. In our case, the entire corpus' parsing (45M sentences) take about a day to complete. Regarding the model, we implemented tree models using a novel batching method, which allows us to keep training in a reasonable range (maximum 41 hours on a Nvidia 1080 Ti.)

[1] Pitfalls in the Evaluation of Sentence Embeddings. Steffen Eger, Andreas Rücklé, Iryna Gurevych. RepL4NLP-2019
[2] Wieting et al., NO TRAINING REQUIRED: EXPLORING RANDOM ENCODERS FOR SENTENCE CLASSIFICATION. ICLR 2019.
[3] https://github.com/yzhangcs/parser
[4] https://spacy.io/models

---

### Decision · Program_Chairs · 2021-01-07
**Final Decision**

**Decision:**

Reject

**Comment:**

I thank the authors both for going the extra mile in doing further experiments for their response, and making the efforts to synthesize the main comments and concerns of the reviewers.

Overall, I'm pretty sympathetic to the idea that syntactic and semantic representations should be very helpful to learning sentence embeddings. They provide a form of scaffolding. But a reviewer notes and I think anyone will admit in 2020 that contextual language models like BERT also provide much of this scaffolding, and it falls to the paper author to provide convincing evidence that using external parsers is valuable and necessary in this quest. In this, the current paper seems to fall somewhat short.

Pros:
 - Clearly and honestly written paper
 - Good exploration of value of constituency & dependency parse representations
 - Exploits recent work in contrastive learning

Cons:
 - Insufficient novelty
 - Experimental comparisons not well controlled – too much apples and oranges.
 - No comparisons of inference speed tradeoffs
 - Value of exploiting explicit syntax is too much assumed rather than explored
 - It's not established that use of explicit syntax really delivers versus alternatives such as contextual language models

Several of the reviewers felt that this paper was a fairly limited extension of L & L 2018, without any clearly novel contribution. The issue of comparability in results is complicated. There is a reason to move to a new standard corpus, rather than privileged people passing around archived copies of the old BooksCorpus, and I think your additional experiments show the results are "near enough" but there would still be much more archival value in a new paper having a set of comparable results on a new corpus. The big question of whether to do this or use BERT is better addressed in your additional experiments presenting a random projection of BERT to a comparable higher dimensional space. But unfortunately these results further weaken the clarity of the case for needing to head in the direction of this paper rather than just using a large pre-trained contextual LM.